# A Deep-Learning-Based Method for Optical Transmission Link Assessment Applied to Optical Clock Comparisons

**Sibo Gui** [1], **Meng Shi** [1], **Zhaolong Li** [1], **Haitao Wu** [1], **Quansheng Ren** [1] and **Jianye Zhao** [1,2,*]

1   School of Electronics, Peking University, Beijing 100871, China; gsb530@stu.pku.edu.cn (S.G.); qsren@pku.edu.cn (Q.R.)
2   Zhongkeqidi Academician Workstation, Guangzhou 510535, China
*   Correspondence: zhaojianye@pku.edu.cn

**Abstract:** We apply the Empirical Mode Decomposition (EMD) algorithm and the Time Convolutional Network (TCN) structure, predicated on Convolutional Neural Networks, to successfully enable feature extraction within high-precision optical time-frequency signals, and provide effective identification and alerts for abnormal link states. Experimental validation confirms that the proposed method not only delivers an efficacy on par with traditional manual techniques, but also excels in swiftly identifying anomalies that typically elude conventional approaches. This investigation furnishes novel theoretical backing and forecasting tools for high-precision optical transmission.

**Keywords:** optical clock; time-frequency transfer; artificial neural network

## 1. Introduction

Optical clocks, as the most precise time-frequency standards [1], are now widely produced by numerous laboratories [2]. This mass production has sparked a key discussion: establishing optical clock networks to compare performance between different optical clocks [3–5]. It is crucial to ensure that the main source of signal jitter during signal comparison or distribution originates from the frequency standard itself, not the optical link transmission process [6].

One common approach involves creating a loop between the local and remote ends [7]. Here, the local reference signal is compared with the signal returning from the remote end, helping to isolate the noise within the optical fiber link [8]. While this method is well-established and broadly used, it becomes increasingly complex with the growth of optical fiber networks and the rise in remote users. The complexity arises from two factors. First, each transmission link utilizes an additional channel within the optical fiber [9], a resource that is limited. Second, local phase detection needs extra optical/electrical equipment [10], suggesting the impossibility of infinite expansion for the optical fiber network. In order to save resources required for real-time monitoring, we started using pre-supervised or unsupervised methods to manage optic fiber networks in the distribution and comparison of optical clock signals. However, the lack of real-time supervision directly weakens the validity of experiments. What makes things worse is that even with the introduction of fiber optic links, there is very little work mentioning real-time evaluation of optic fiber links, which means that we cannot immediately respond to link anomalies. These two situations not only make it inevitable to compare the complex optical clock signal between three locations, but also greatly increase the difficulty of putting the optical clock signal into practical use. This situation necessitates the exploration of novel techniques, enabling the direct and real-time evaluation of frequency transmission quality in the optical fiber link using received time-frequency signals at the remote end.

The primary goal of evaluation is to separate the noise generated by the optical fiber link from the noise produced by the clock. Our current understanding of the non-linear effect of light within the optical fiber is still insufficient, and we still cannot establish a

noise model for the optical link transmission process. Consequently, it is challenging to fully differentiate these two noise types. So, the primary task becomes finding suitable data features for classification. The slope of Allan variance plots, derived from the power spectra of noise, is a standard method for identifying noise sources. Notably, both ultra-stable lasers and optical combs exhibit different characteristics from optical fibers over longer average times. Moreover, it's been suggested that the primary noise in the optical fiber link is high-frequency noise, thus incorporating frequency into consideration [11].

Modern advancements in signal processing are providing an array of unsupervised mode decomposition and adaptive feature extraction methods [12], particularly those based on deep learning techniques within neural networks [13]. These methods can construct adaptive filters via complex parameters. Successful integration of mode decomposition with neural networks has produced superior outcomes in classification and assessment task based on weather [14], traffic [15], and power [16] data.

Noise is often sampled at regular intervals, and thus in the field of data processing, it can be considered as a time series. A time series is a sequence of data points arranged in the order of their occurrence over time. Typically, the time interval for a set of time series is a constant value (such as 1 s, 5 min, 12 h, 7 days, 1 year), so the time series can be analyzed and processed as discrete time data. Time series assessment is a basic task based on time series, with the aim to identify abnormal events or behaviors from normal time series. It can detect from historical data, and can also provide early warnings for abnormities that have not yet occurred based on time series assessments. The basic method is to decompose the time series to form multiple features, and then classify using artificial neural networks. The WISDM laboratory first proposed the task of human activities recognition (HAR) using mobile sensor data [17], which defined the task of time series prediction for the first time, and provided a reference for time series data processing methods. Since then, a large number of time series assessment methods based on the HAR dataset have been proposed, such as CNN-LSTM [18], ConvLSTM [19], and DeepConvLSTM [20]. The structure based on convolutional neural networks and recurrent neural networks has formed the prototype method for time series prediction tasks.

In this study, we introduce mode decomposition methods to extract low-frequency noise caused by temperature and vibration in optical fiber link noise. Subsequently, we adopt neural network algorithms to assess the stability of the optical fiber link in optical clock comparison experiments. Our results demonstrate that our proposed method provides a direct, accurate, and rapid assessment of the optical fiber link's transmission performance at the remote end.

## 2. Materials and Methods

This study presents a feature extraction and state prediction model that can be utilized for optical clock signal distribution and comparison. We will sequentially introduce the data source for the model, the feature extraction methodology, and the state assessment approach in this chapter.

### 2.1. Data Source

Our data were collected from several different laboratories, as we aimed to ensure the universality of our algorithm. The measurements of optical carrier microwave transmission noise based on fiber Sagnac-loop-based optical-microwave phase detectors (OM-PD), conducted by Jung et al. in 2014 [10] and Leng et al. in 2016 [21], provided us with typical link noise values under conditions that met the requirements for optical clock transmission. By applying an all optical link to achieve phase locking and phase discrimination, OM-PD avoids the impact of introducing electrical noise during the measurement process. We used these data to identify appropriate mode decomposition methods. The data we used for simulating the comparison of optical clocks in a real experiment scenario comes from Scioppo et al. in 2022 [11]. While they provide data collected in normal transmission conditions, we manually measured the noise data when faced with abnormal transmission

link conditions. These included inadaptable feedback coefficients, sudden data hopping, and exposure to excessive vibrations. Based on these data, we created link noise dataset that concludes data under either conditions that do or do not meet the requirements for optical clock transmission. We used the dataset mentioned above to observe the model's assessing capabilities.

### 2.2. Signal Processing and Feature Extraction

In our research, the link noise was sampled as a discrete time series signal at 1 Hz. Many important investigations of this noise signal revealed some important findings: firstly, environmental temperature can affect the length and refractive index of optical fibers, thereby introducing noise [22]. This noise should exhibit long-period characteristics that coincide with temperature changes. Secondly, sound and infrasound waves with frequencies between $10^{-4}$ Hz and 1 kHz can introduce periodic noise by affecting the length of optical fibers [23]. Finally, electrical devices can also introduce high-frequency electrical noise.

Considering our sampling rate is only 1 Hz, we specifically took the Nyquist sampling theorem into account. The Nyquist sampling theorem implies that high-frequency noise with a frequency higher than 0.5 Hz not only cannot be restored but also contaminates the low-frequency spectrum. Therefore, the algorithm's goal is to extract two types of low-frequency noise mentioned above, while filtering out the high-frequency noise from the signal. The structure of our signal processing module is shown in Figure 1, which includes smoothing, wavelet transformation, and empirical mode decomposition.

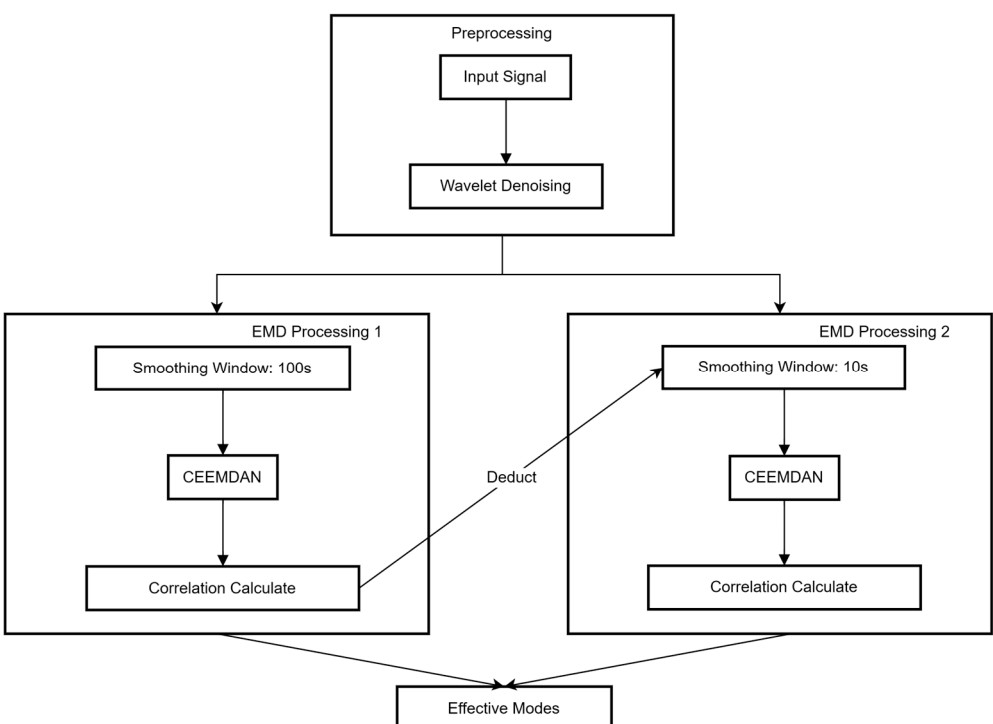

**Figure 1.** The structure of signal processing and feature extraction algorithm.

We used wavelet threshold denoising [24], a common method in the signal processing field to remove noise, which can effectively filter out high-frequency white noise. Simultaneously, we used sliding windows of various sizes to minimize the impact of high-frequency electronic noise. This was due to our observation that the main constituents of high-frequency electronic noise—thermal noise and granular noise—both exhibit zero mean.

The core part of our algorithm is the Empirical Mode Decomposition (EMD). EMD is a method designed for the analysis of nonlinear and non-stationary time series data. Introduced by Norden E. Huang and his team in 1998, this technique decomposes a complex signal into a series of Intrinsic Mode Functions (IMFs). These IMFs represent oscillatory modes embedded within the original signal, each characterized by a specific frequency, amplitude, and phase, which may vary over time. The EMD method is adaptive and data-driven, requiring no a priori basis, and is particularly effective for analyzing signals that exhibit temporal variations in their spectral properties. By using EMD and its series of algorithms, we can separate the main components in the signal, which have different frequency. Thus, we can use this technique to separate noise introduced by infrasound and temperature, which have distinct differences in frequency domain. However, due to the contamination caused by low-frequency noise and severe endpoint effects, traditional EMD algorithm may lead to severe mode mixing in main components. To solve this issue, we used the improved Complementary Ensemble Empirical Mode Decomposition with the Adaptive Noise algorithm (CEEMDAN). By adding white noise to the original signal multiple times, this algorithm performs EMD on each of these noisy versions of the original signal, and then takes an ensemble average to obtain the final result. It has been proved that CEEMDAN can precisely decompose signal into several main components with different frequency domains [25].

To confirm which main components belong to effective modes, we calculated the correlation between the output components from the algorithm and the original signal and retained those with higher correlation as effective modes.

Given that the decomposition process of empirical mode decomposition is mainly based on empirical data rather than definitive physical models, we will attempt to provide a physical interpretation of these effective modes to demonstrate the effectiveness of the decomposition. Then, we will combine these effective modes as features, and serve as inputs for the final assessment task.

### 2.3. Data Classification and State Assessment

After obtaining sufficient features, our task becomes utilizing these features to perform link state assessment. Given that the features are discrete time series, this can be viewed as a time series classification problem. In recent years, the topic of time series prediction/classification has been widely studied, with various structures based on Recurrent Neural Networks (RNN) showing impressive performance. However, due to the sequential nature of the calculations in RNN, a large number of activation vectors needed for intermediate calculations must be stored during training and predicting [26], requiring much more memory than CNN. Here, we used the Temporal Convolutional Networks (TCN) structure based on CNN to classify discrete-time series signals [27].

The key operations in TCN is using dilated causal convolution to extract features from extensive historical information. Causal means the convolution result at any given moment will only come from the convolution result of the data from previous moments. At the same time, we introduced a dilatation factor to extract features from longer historical information without substantially increasing the network layers of the neural network. The TCN we used contains four temporal blocks, each performing two convolution operations and one residual operation. The specific operations contained in a temporal block are presented in Figure 2.

The time series input to temporal block first goes through the dilated convolution layer, then through a non-linear activation function layer and a dropout layer, then into the second dilated convolution layer, and again through a non-linear activation function layer and dropout layer. The result is subtracted from the original data as a residual, yielding the result of this layer. To keep the network stable, we added a batch-normalized operation after each dilated convolution layer.

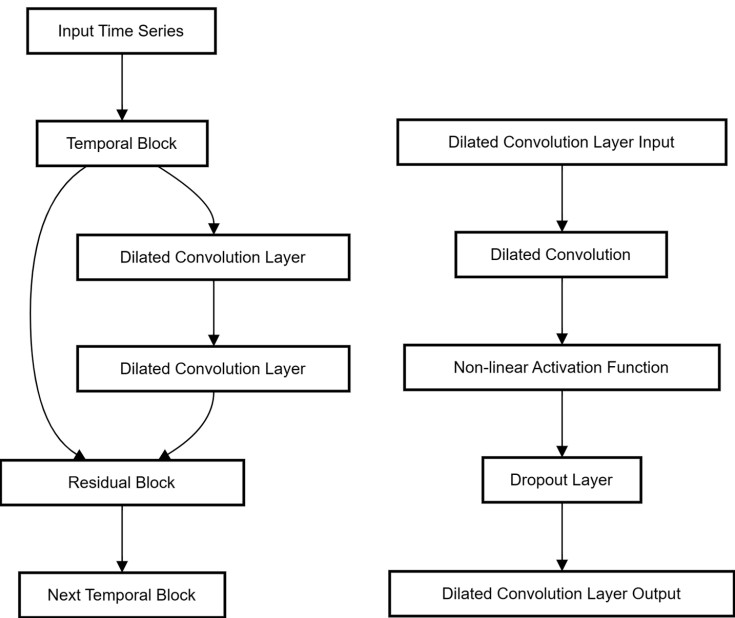

**Figure 2.** The structure of temporal block (**left**) and dilated convolution layer (**right**).

## 3. Results

### *3.1. Feature Extraction*

Given the extremely wide range of signal frequency bands, we used two mode decomposition structures with different smoothing window lengths. In this chapter, we will sequentially demonstrate the capabilities of the two structures in decomposing effective components and extracting effective features, and provide a physical interpretation for these effective features.

#### 3.1.1. Filter 1: With 100 s Smoothing Window

Using a 100 s smoothing window and EMD processing, the effective mode mainly obtained components whose periods are over 2000 s. It is shown with the original data in Figure 3. As seen, the principle components decomposed from Filter 1, which mean effective mode (red line) reflects the variation of the original data (green line) over longer periods.

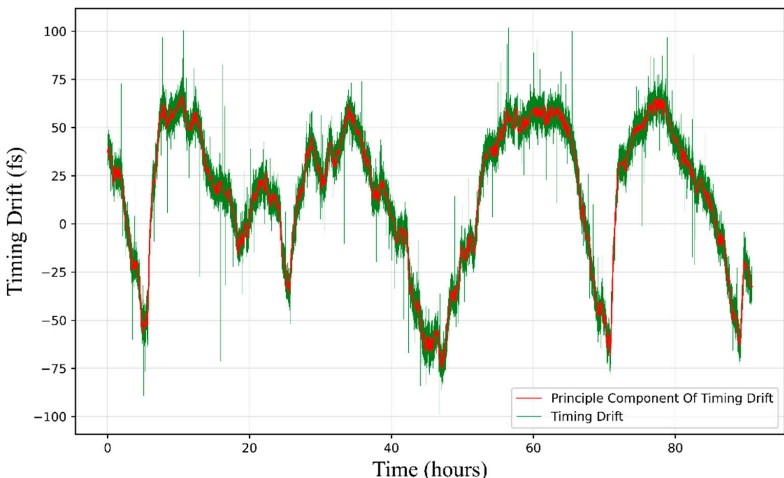

**Figure 3.** The principle component decomposed from Filter 1, and the original data graph, The red line represents features and the green line represents the original data. Original data from Jung et al. [10]. We use data spanning 93 h to demonstrate the algorithm's effectiveness in extracting long-time effects caused by temperature.

Figure 4 reflects the relationship between the principle components and laboratory temperature. It can be seen that the principle components largely reflect the impact of laboratory temperature, which means it can be seen as a feature representing temperature. Other high-frequency fluctuations in the feature may be due to low-frequency infrasound waves or the integration of high-frequency noise with a non-zero mean.

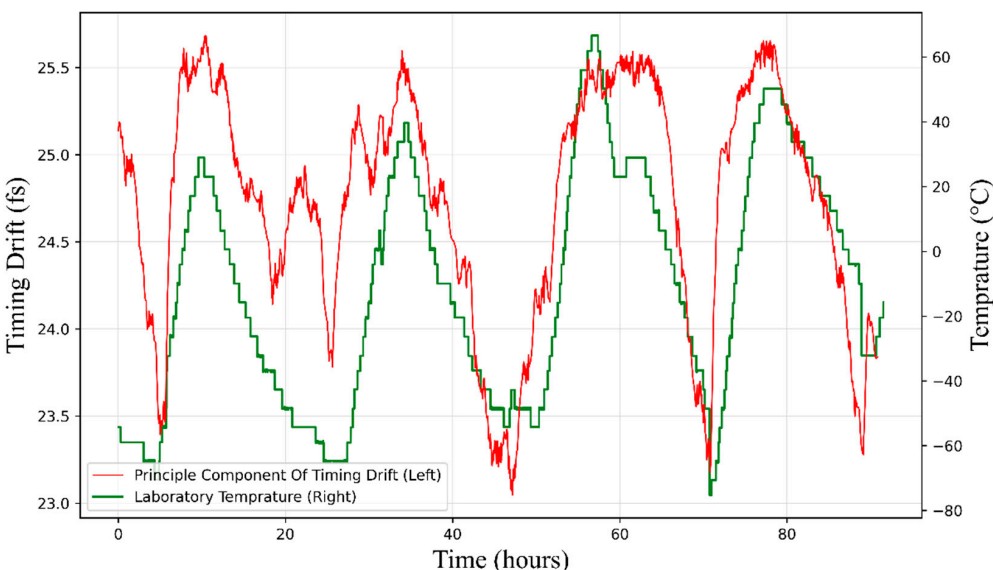

**Figure 4.** The relationship between the temperature feature and real laboratory temperature. The red line represents the feature. The green line represents laboratory temperature. Original data from Jung et al. [10].

### 3.1.2. Filter 2: With 10 s Smoothing Window

At this point, we smoothed the noise using a 10 s smoothing window, subtracted the temperature feature already extracted above, and then performed EMD processing. The relationship between the principle components obtained and the remaining signal is shown in Figure 5. It can be seen that the selected components can follow the remaining noise changes well.

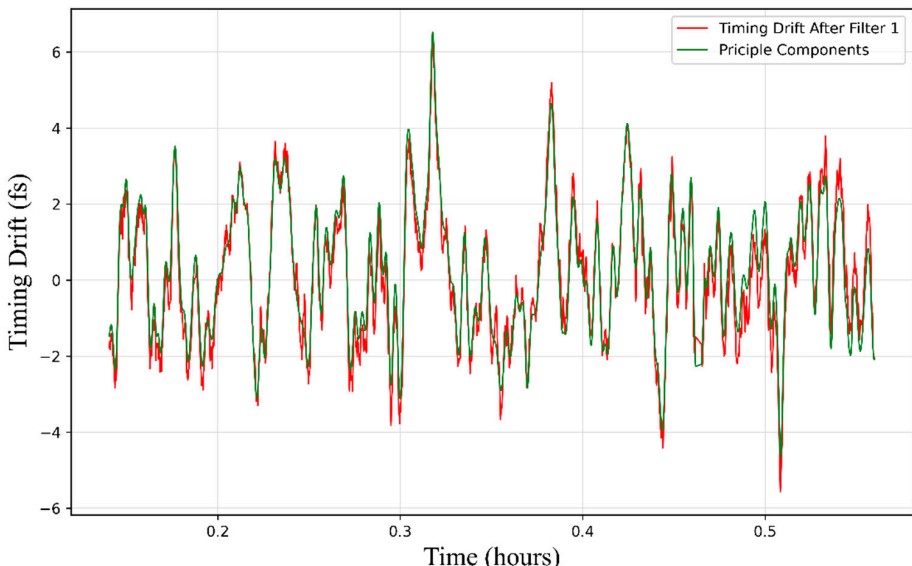

**Figure 5.** The principle components extracted from the second EMD operation, and the remaining data graph. The red line represents the components. The green line represents remaining data after extraction from filter 1.

Unlike the structure above, the effective mode selected in the second structure is composed of main components of different frequencies. The specific situation is shown in Figure 6. We infer that the main components of the signal are caused by infrasound waves with periods ranging from 10 s to 1000 s, which means the principle components can be seen as a feature representing infrasound.

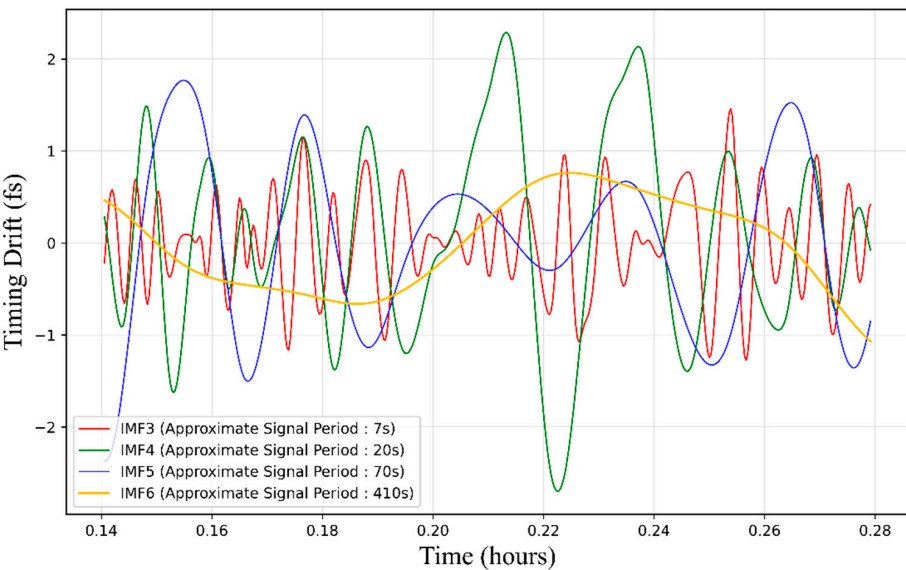

**Figure 6.** The main components obtained from the second structure. Lower left corner indicates the estimated periods of the main components in four different frequency bands.

### 3.2. State Assessment

In this section, our main task is to verify whether the neural network based on the features above can assess the link status and alert when anomalies occur. Firstly, we mixed the data mentioned in Section 2.1 to construct noise signals when the link status was normal or abnormal. Then, we used the feature extraction algorithm proposed in the last chapter to extract effective features. Finally, we constructed the test set and training set for the training model.

Based on the experience of Section 3.1, it was not difficult to find that the effective features we extracted are mainly distributed over a scale of tens to thousands of seconds. Therefore, we cut out the time series of 2000 s duration from the noise signal as features for prediction, and made training and testing sets; one fifth of total data was used for testing. To ensure that the model learns the shape features of the time series rather than the size, we normalized the time series in the training set in advance.

The effective features extracted from normal signals and abnormal signals are shown in Figure 7. We found that the extraction result corresponding to the normal link state is a combination of several mid-frequency noises, which is consistent with our analysis in Section 3.1. Abnormal signals behave different from normal ones. For example, when the feedback link fails (with inadaptable feedback coefficient), a large amount of high-frequency noise immediately appears in the extracted features, indicating that the high-frequency component ratio in link noise increases at this time, thus being extracted into the features. Sudden data hopping corresponds to significant changes over a long period, so the impact of these changes on the extracted features far exceeds that of other noise. This means that the features of link noise in these abnormal states are different from the normal state, which gives the model the ability to predict through learning.

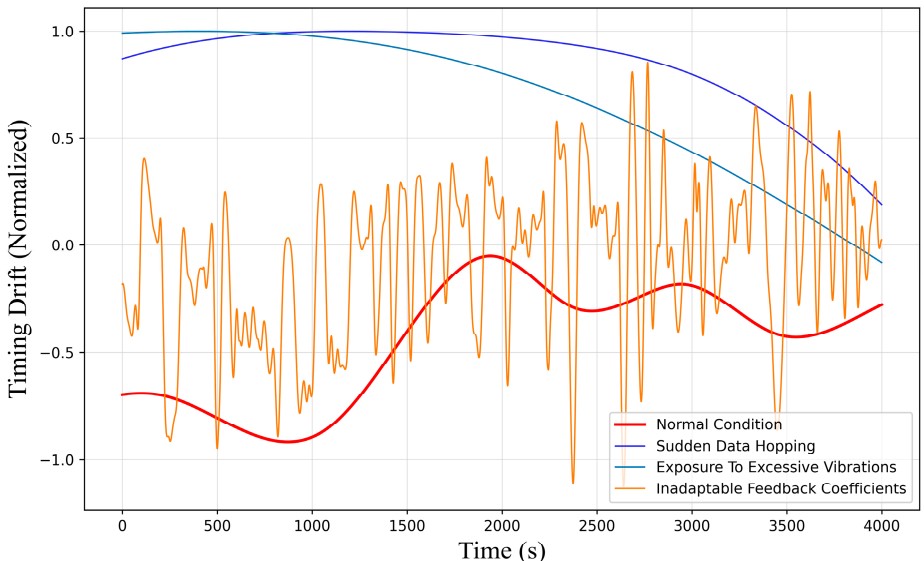

**Figure 7.** The effective features extracted from normal signals and abnormal signals.

We hope that the assessment model based on neural networks can alert when the link state is abnormal. Because the signal is continuous, we can continuously identify abnormal states in real applications. Obviously, false negatives (recognizing normal states as abnormal) are less desirable than false positives (recognizing abnormal states as normal), so we adjusted the way the loss metric was written during neural network training to make the proportion of false negatives as low as possible. After our careful selection, the accuracy, precision, recall, and F1 score in the test dataset are shown in Table 1. Accuracy in the test dataset was 71.32%, with the Recall Rate of normal data up to 91.71%; it minimizes the interference of false negative samples as much as possible.

**Table 1.** The accuracy, recall, and F1 score of the prediction task in test dataset.

| Label | Precision | Recall | F1 Score |
| --- | --- | --- | --- |
| Normal | 60.66% | 91.71% | 73.18% |
| Abnormal | 90.27% | 56.43% | 69.25% |

In the previous section, we discussed the performance of the model on the test dataset, which contains three different types of abnormal samples, and we are concerned with the overall performance of the model. Next, we will delve further into the effectiveness of the model in dealing with real anomalies. We assume that the transmission link runs normally for the first 4000 s, starts to experience anomalies from 4001 s, and continues until 7500 s. We use this as validation data to validate the actual performance of the model. As Table 2 shown, the accuracy on the validation set is consistently above 80%, which proves our model can accurately assess the link state.

**Table 2.** The accuracy of the prediction task in validation dataset.

| Status | Accuracy |
| --- | --- |
| inadaptable feedback coefficients | 91% |
| exposure to excessive vibrations | 83% |
| sudden data hopping | 81% |

## 4. Discussion

In preceding research endeavors, there has been a consistent lack of methods for directly evaluating optical fiber transmission links. Common practices involve first ensuring

that the link can operate with low noise, and then manually eliminate experimental data that appears to have abnormal link conditions in long-term experiments. This situation has been established in many experiments involving optical clock comparison and distribution. If we can prove in advance that the link can operate stably, the optical clock comparison completed with this link might be valid, but because of the lack of continuous assessment through the whole experiment, it cannot be proved. That means the experiment is still incomplete, and the results are unconfirmed. This makes complex optical clock trilateral comparisons inevitable, and also makes evaluation for the distribution of optical clock signals exceptionally complicated. The method proposed in this study introduces, for the first time, an automated real-time assessment mechanism. Next, we will explore the effectiveness and applicable scenarios of the method in sequence.

### 4.1. Effectiveness

In the following sections, we will explore through illustrative examples whether the proposed method can provide a supplementary link assessment method, assisting us in assessing the state of optical fiber links and enhancing the validity of optical clock comparison and distribution experiments.

Common anomalies include a sudden increase in link noise, escalating to tens of femtoseconds. Manual methods can provide warnings by observing the data of link noise and sliding window Allan variance. We will first demonstrate that the method proposed in this paper can achieve similar effects. We will use a sample exposed to excessive vibration as an example. We found that the model started to output continuous failure alerts after 220 s, achieving an assessment effect similar to manual methods. The original signal and the Allan variance with a sliding window of 1000 s are shown in Figure 8.

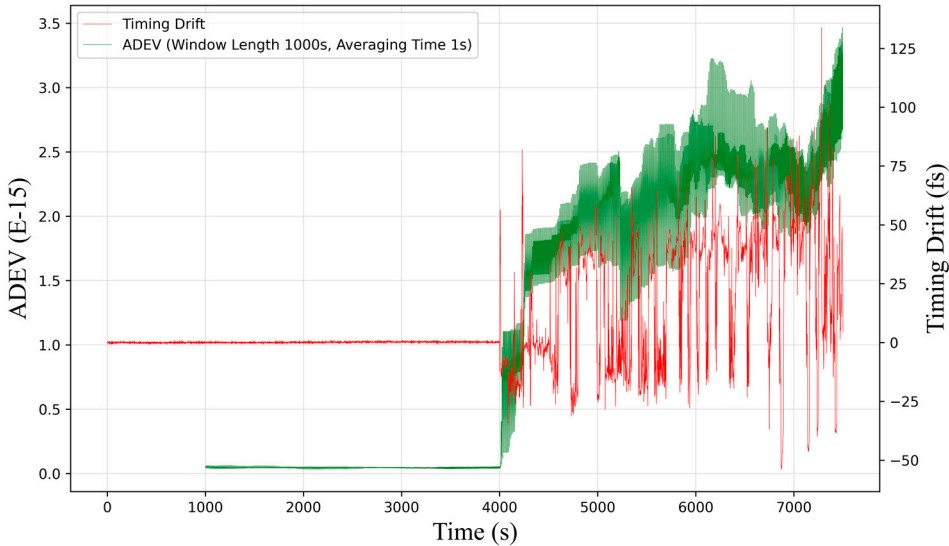

**Figure 8.** The original signal at this point and the Allan variance with a sliding window of 1000 s. Vibration occurs in 4000 s.

When the strength of abnormal signals is weak, relying solely on observation may often be insufficient to identify anomalies. We will use another sample exposed to vibration as an example. The original signal at this point and the Allan variance with a sliding window of 1000 s are depicted in Figure 9. Under these conditions, the model identifies link failure after 1319 s and continuously outputs failure alerts. It can be observed that after vibration occurs in 4000 s, the fluctuations of the noise signal are not clearly apparent, making it challenging for traditional methods to identify it. This implies that the model proposed in this paper has the ability to accurately identify anomalies in the link.

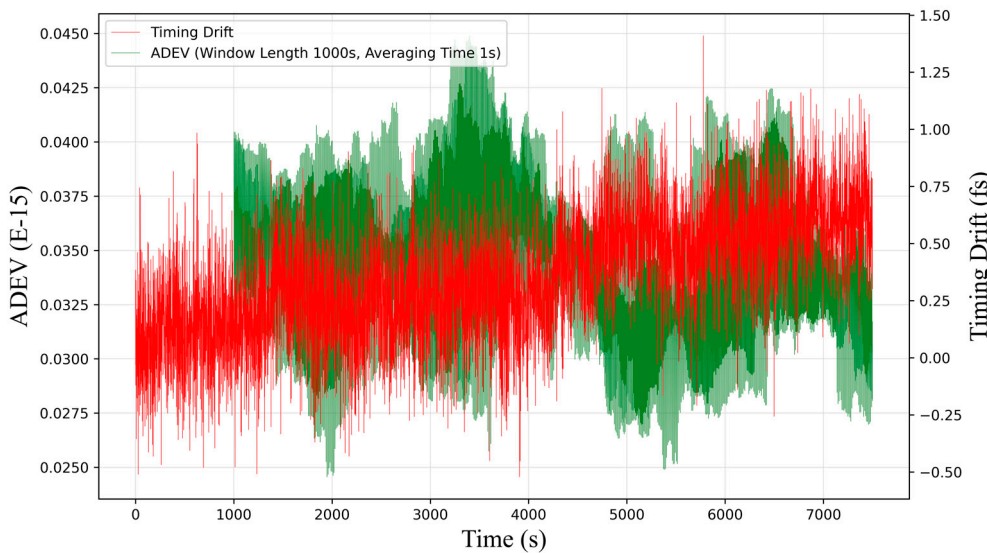

**Figure 9.** The original signal at this point and the Allan variance with a sliding window of 1000 s. Vibration occurs in 4000 s.

The assessing speeds for three anomalies are also quite rapid, specifically 720 s for exposure to excessive vibrations, 394 s for inadaptable feedback coefficients and 824 s for sudden data hopping. This indicates that our model also has the capability of fast alert, indicating potential anomalies promptly.

### 4.2. Applicable Scenerios

Considering that in current optical clock comparison experiments it is common to verify the link state through feedback or loop verification before conducting experiments, the method shown in this article can be easily integrated into the current experiment. The specific steps can be summarized as the following three steps: Firstly, collect enough link noise signals and manually mark whether there are anomalies; Secondly, use the clock compared signal and link noise signal to jointly construct data set, and train the model; Finally, use the model directly in optical clock comparison experiment. This means that our method has certain practicality for current light clock comparison experiments.

We further test the generalization ability of the model when the training data comes from several different links. We collected normal and abnormal data using another fiber optic link and established a new dataset. In order to compare the performance of the model, 50% of the training data in the new task comes from the new dataset, while all the test data still comes from the dataset used in the previous section. While maintaining the size of the training set unchanged, the accuracy of the model decreased to 67.50%. On the one hand, this performance can still ensure the effectiveness of the model. On the other hand, a performance loss of approximately 4% reminds us that changes in sample distribution can damage the performance of the model, and the generalization ability of the model still needs to be improved.

## 5. Conclusions

In conclusion, this study has presented a novel approach to the assessment of optical transmission links, leveraging the power of deep learning and signal processing techniques. The proposed method, which combines the Empirical Mode Decomposition algorithm and the Time Convolutional Network structure, has proven effective in extracting meaningful features from high-precision optical time-frequency signals and providing real-time alerts for abnormal link states. This advancement not only enhances the validity of optical clock comparison and distribution experiments but also paves the way for more efficient and reliable optical transmission.

Considering that neural networks are introduced into this niche field for the first time, this study did not perform specific optimizations for the task at hand. The sequence decomposition and prediction methods used are built on established models, suggesting that there is considerable potential for further exploration of neural network-based link state assessments. A significant limitation of this study is the lack of diversity in data sources, which implies that the generalizability of the method remains open to question. We sincerely welcome other researchers to attempt to reproduce the results of this study in other experimental scenarios. Further information about our algorithm can be seen in Appendix A.

Future work can focus on three directions: 1. Hardware implementation: The method proposed in this study could be implemented on hardware, truly integrating it into the system of optical clock distribution. 2. Precision enhancement: The prediction accuracy of the network used in this study is not high enough. New network structures and detailed decomposition methods may further improve network accuracy. 3. Network application: In the actual application scenario of optical clock distribution, due to more clock sources and more links, we can receive more valid signals. We might be able to use other signals to predict the quality of a link's transmission.

We believe that research in these three areas can further broaden the application scenarios of the results of this study, truly applying new methods such as neural networks to optical clock comparison and distribution.

**Author Contributions:** Methodology, J.Z. and S.G.; Algorithm, M.S.; Validation, S.G. and M.S.; Data Resources, Z.L., S.G. and H.W.; writing—original draft preparation, S.G.; writing—review and editing, J.Z. and Q.R.; funding acquisition, J.Z. and Q.R. All authors have read and agreed to the published version of the manuscript.

**Funding:** This work was supported in part by National Key Research and Development Program of China (2021YFB2801900), National Natural Science Foundation of China (91836301).

**Institutional Review Board Statement:** Not applicable.

**Informed Consent Statement:** Not applicable.

**Data Availability Statement:** The code used in this paper can be obtained by contacting the corresponding author.

**Acknowledgments:** Supported by High-performance Computing Platform of Peking University.

**Conflicts of Interest:** The funders had no role in the design of the study; in the collection, analyses, or interpretation of data; in the writing of the manuscript; or in the decision to publish the results.

## Appendix A

The software part of this article is written in Python. The implementation of the data processing part is based on two packages, PyWavelets [28] and PyEMD [29], which provide convenient implementations of wavelet denoising and empirical mode decomposition, respectively. The specific installation steps and usage methods can be found on the website provided by the publisher mentioned in the comments.

Our implementation of neural networks is based on the PyTorch framework. First, we built the Temporal Block module, and then formed the TCN network by stacking this module. In the task performed in Section 3.2 of this article, we used a four-layer TCN network with a deletion factor of 4, a batch size of 64, a dropout rate of 0.2, and a number of epochs of 100, and recorded the experimental results shown in the text. We can almost confirm that the setting of TCN layers and Receptive field is closely related to task objectives, number of epochs, batch size and other super parameters, so it may be meaningless to copy the super parameters in this paper.

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
