# Peer review of "A Deep-Learning-Based Method for Optical Transmission Link Assessment Applied to Optical Clock Comparisons"

_photonics, doi:10.3390/photonics10080920_

Round 1

Reviewer 1 Report

The research paper brings up an interesting discussion of ways to characterise optical fibre links using machine learning methods. The novel approach could be especially useful for long fibre links where it is hard to find the cause of the problem or for long measurement campaigns where it is necessary to quickly identify the problem.

1) The authors argue that commonly used fibre loop techniques are becoming increasingly complex with the growing scale of optical fibre networks and require attention during experiments. However, the introduction briefly misleads that the developing approach could be used instead of the fibre loop. It becomes apparent later that, first, the new approach could be used for characterising long-term (>> 1s) fluctuations/noises, and second, only for assessment of the link (as of yet). I think it is better to clearly state that, focusing on advantages of using this method together with the commonly used fibre loops.

2) The analysis of the method performance could be improved. It is important to know about differences between models trained using different scenarios and their accuracy of fibre link assessment:

a) model trained on only 1 fibre link, used for that link only

b) model trained on only 1 fibre link, used for another link

c) model trained on many links, used for any of those links

d) model trained on many links, used for another link

e) impact of lengths of the fibre links

I think it is important to include in the discussion at least some analysis that might be useful for implementation in real experiments without additional research.

Apart from these issues, I think the paper is well written and is suited for publication. I advise publication of the code used in this paper in an open-source repository.

Reviewer 2 Report

The authors present a deep-learning-based method for optical link anomaly detection. I have a few comments.

1. Acronyms. (i) Line 83, what's "OM-PD"? (ii) Line 131, what's the "CEEMDAN" algorithm? Can you briefly describe it?

2. What software libraries/packages did you use to achieve EMD and TCN? It's good to provide detailed information in Appendix so that people can possibly replicate your result.

3. In the Discussion section, you compared your anomaly-detection algorithm with the conventional Allan-variance algorithm, which is a good study. However, in my opinion, even a simple algorithm can detect anomalies in Figures 8 & 9. For example, I'd like to propose this algorithm "multi-scale error detection" -- we sample the data shown by the red curve every 1 second, 2 seconds, 4 seconds, ... , 256 seconds; then we have a sliding window of 10 points for each sampling interval; next we compute the standard deviation (STD) of those 10 points for each sampling interval and if one of the STDs is larger than a pre-known threshold then we find an anomaly. It seems to me that this simple algorithm is intuitive and easy to implement, and does not involve the "black-box" feeling in deep learning. Can you compare your deep learning method with this "multi-scale error detection" algorithm and see how well the deep learning method performs? 

4. For reference 4, it's better to cite this one "Frequency ratio measurements at 18-digit accuracy using an optical clock network," Nature 591 (7851), 564-569, 2021.
